# Loss of Expression of Antiangiogenic Protein FKBPL in Endometrioid Endometrial Carcinoma: Implications for Clinical Practice

**DOI:** 10.3390/medicina58101330

**Published:** 2022-09-22

**Authors:** Danilo D. Obradović, Nataša M. Milić, Nenad Miladinović, Lana McClements, Dejan M. Oprić

**Affiliations:** 1Institute of Pathology, Faculty of Medicine, University of Belgrade, 11000 Belgrade, Serbia; 2Institute for Medical Statistics and Informatics, Faculty of Medicine, University of Belgrade, 11000 Belgrade, Serbia; 3Department of Pathology, Clinical Hospital Centre Zemun, 11000 Belgrade, Serbia; 4Faculty of Science, School of Life Sciences, University of Technology Sydney, Sydney, NSW 2007, Australia

**Keywords:** FKBPL, endometrioid endometrial carcinoma, angiogenesis, VEGF-A, estrogen receptor alpha

## Abstract

*Background and Objectives*: FK506 binding protein like (FKBPL) is a member of the immunophilin family, with anti-angiogenic effects capable of inhibiting the migration of endothelial cells and blood vessel formation. Its role as an inhibitor of tumor growth and angiogenesis has previously been shown in studies with breast and ovarian cancer. The role of FKBPL in angiogenesis, growth, and carcinogenesis of endometrioid endometrial carcinoma (EEC) is still largely unknown. The aim of this study was to examine the expression of FKBPL in EEC and benign endometrial hyperplasia (BEH) and its correlation with the expression of vascular endothelial factor-A (VEGF-A) and estrogen receptor alpha (ERα). *Materials and Methods*: Specimens from 89 patients with EEC and 40 patients with BEH, as well as histological, clinical, and demographic data, were obtained from the Clinical Hospital Centre Zemun, Belgrade, Serbia over a 10-year period (2010–2020). Immunohistochemical staining of the tissue was performed for FKBPL, VEGF-A, and ERα. Slides were analyzed blind by two pathologists, who measured the intensity of FKBPL and VEGF-A expression and used the Allred score to determine the level of ERα expression. *Results*: Immunohistochemical analysis showed moderate to high intensity of FKBPL expression in 97.5% (*n* = 39) of samples of BEH, and low or no expression in 93.3% (*n* = 83) of cases of EEC. FKBPL staining showed a high positive predictive value (98.8%) and a high negative predictive value for malignant diagnosis (86.7%). The difference in FKBPL expression between EEC and BEH was statistically significant (*p* < 0.001), showing a decrease in intensity and loss of expression in malignant tissues of the endometrium. FKBPL expression was positively correlated with ERα expression (intensity, percentage and high Allred score values) and negatively correlated with the expression of VEGF-A (*p* < 0.05 for all). *Conclusions*: FKBPL protein expression demonstrated a significant decrease in FKBPL in EEC in comparison to BEH tissue, with a high predictive value for malignancy. FKBPL might be emerging as a significant protein with antiangiogenic and antineoplastic effects, showing great promise for the diagnostic and therapeutic applications of its therapeutic derivatives in gynecological oncology.

## 1. Introduction

Endometrial carcinoma (EC) is the most common gynecological malignant tumor and the fourth most common malignancy in women in the United States, with a rising incidence and mortality rate [1]. The highest incidence of EC is observed in post-menopausal women, although nearly 20% of women are diagnosed before menopause, and approximately 5% of women are diagnosed before the age of 40. In addition to EC, the most frequently found hyperplastic change of the endometrium is endometrial hyperplasia without atypia/benign endometrial hyperplasia (BEH). BEH is a hormonally induced change of the endometrium with a 1–3% risk of developing EC. The main risk factors for the development of EC are high levels of circulating estrogens, exposure to exogenous estrogens, obesity, late menopause, nulliparity, a history of polycystic ovary syndrome, tamoxifen therapy, and Lynch’s syndrome. Histologically, EC is classified as endometrioid, serous, clear cell, undifferentiated, dedifferentiated, mixed carcinoma, and carcinosarcoma of the uterine corpus [2,3,4,5,6]. Endometrioid endometrial carcinoma (EEC) accounts for 85% of cases. EEC is a highly estrogen-dependent tumor, with a plethora of evidence supporting estrogen stimulation unopposed by progesterone as one of the central mechanisms in its carcinogenesis. Estrogen receptor alpha (ERα) is the main factor through which estrogens stimulate mitogenic and proliferative activity in healthy endometrium and EEC [7]. It is shown that estradiol (E2) stimulates the proliferation and migration of endometrial cancer stem cells (CSC) through ERα. Paradoxically, the presence of ERα positivity in EECs is shown to be a positive prognostic marker, as opposed to ERα negative EECs that are more aggressive tumors with poorer prognoses, which might be due to the dedifferentiation of tumor cells [8].

The growth of EEC, before the tumor exceeds a volume of 2 mm^3^, is independent of vascularization, obtaining nutrients and oxygen by diffusion, but for further growth formation of new blood vessels, it is necessary to support the metabolic requirements of neoplastic tissue [9]. Blood vessel growth occurs in the form of vasculogenesis and angiogenesis. Vasculogenesis occurs via the differentiation of angioblasts, and it is characteristic of embryonal development, while angiogenesis is growth from previously existing blood vessels. Angiogenesis is a physiologically highly regulated process, mainly stimulated by tissue hypoxia. Tumor growth is characterized by a loss of control over angiogenesis by the constant production of pro-angiogenic and decrease in anti-angiogenic factors, resulting in the rapid formation of the blood vessels, called the “angiogenic switch” [9].

The vascular endothelial growth factor (VEGF) family is a group of pro-angiogenic ligands that establish their effects through specific receptors. VEGFs have a central role in the process of physiological and pathological angiogenesis. VEGF-A is the most prominent and frequently studied angiogenic factor, shown to have increased expression in EC following the dedifferentiation of tumor cells. Increased VEGF-A expression in EC is a marker of a worse prognosis. Estrogen stimulates angiogenesis through VEGF-A expression in EC [9,10,11].

FK506 binding protein like—FKBPL—is a divergent member of the immunophilin family. It is a well-established anti-angiogenic protein, exhibiting its effects by targeting the cell surface receptor, CD44, on actively migrating endothelial cells, thus inhibiting migration and blood vessel formation. FKBPL’s role as a negative regulator of tumor growth, metastasis, and angiogenesis is established in studies of breast cancer (BC), where FKBPL’s high expression has been associated with a better prognosis of BC. FKBPL and its peptide derivative, ALM201, are shown to decrease the migration and invasion of breast cancer stem cells (CSC) and inhibit the growth of mammospheres of endocrine therapy-resistant breast CSC [12,13]. FKBPL is a part of the HSP90/ERα co-chaperone complex. A stable overexpression of FKBPL in breast CSCs is followed by an increased estrogen dependence on growth and by a higher sensitivity to tamoxifen therapy. A high expression of FKBPL is followed by a decrease in protein levels of ERα in breast CSCs [14]. Studies on ovarian cancer have shown that FKBPL therapeutic peptide derivatives stimulate the differentiation of ovarian CSCs and decrease their numbers while delaying tumor initiation and the growth of highly vascularized xenografts, in conclusion, establishing their antitumor effect through the disruption of angiogenesis [15]. On the other hand, reports on FKBPL’s role in EC are lacking. In a single study examining the genetic profile of EC, it was reported that the expression of FKBPL was observed in stromal cells [16]. According to available data, the expression of FKBPL was found in the epithelium of endometrial glands showing moderate and high intensities of cytoplasmic, membranous, and nuclear expression, with typically pronounced luminal positivity; no expression was reported in stromal cells [17]. In EEC tumor cells, data showed expression varying from none to low and rarely of moderate intensity, whereas the information on FKBPL expression within stromal cells of EEC was not reported [18]. The aim of this study was to examine the expression of FKBPL and its correlation with the expression of VEGFR and ERα in BEH and EEC.

## 2. Materials and Methods

### 2.1. Collection of Samples and Data

Samples were obtained from the Clinical Hospital Centre Zemun, Belgrade, Serbia, over a 10-year period (2010–2020). The study enrolled a total of 89 patients undergoing hysterectomy who were diagnosed with EEC and 40 patients undergoing explorative curettage with a diagnosis of BEH. Inclusion criteria for patients diagnosed with EC included: performed hysterectomy, the availability of clinical staging data, and the availability of paraffin blocks containing tumor tissue covering at least 5 mm^2^. Exclusion criteria included: other histological types of EC, an insufficient amount of tumor tissue for immunohistochemical staining and analysis, other malignant diseases, and previous oncologic therapy. Additional parameters analyzed for patients with EEC were: histological tumor grade, tumor stage (according to Tumor-Node-Metastases (TNM)–based staging and International Federation of Gynecology and Obstetrics (FIGO) classification), depth of myometrial invasion (DMI), and the presence of lymphovascular invasion (LVI). The ethics committee of the Clinical Hospital Centre Zemun approved this study (reference number: 12/1, 29 April 2021).

### 2.2. Immunohistochemical Staining

Tissue sections (4 µm thick) were deparaffinized and dehydrated. Antigen retrieval was performed using Tris-buffer pH 9.0 for FKBPL, and citrate buffer pH 6.0 for VEGF and ER, in a water bath for 30 min at 95 °C. In order to block endogenous peroxidase, slides were treated with a solution containing 3% hydrogen peroxide in phosphate-buffered saline (PBS) for 10 min, and non-specific antigen binding was blocked using a 1% bovine albumin serum (BSA) solution in PBS for 30 min. The primary antibodies used were: rabbit-polyclonal anti-FKBPL (catalog number: 10060-1-AP, Proteintech, 1:800), mouse monoclonal anti-VEGF (VG-1, Santa Cruz, Dallas, TX, USA, 1:100), and mouse monoclonal anti-estrogen receptor α (clone 6F11, Novocastra, Newcastle upon Tyne, UK, 1:100) for a 1 h incubation period at room temperature. After incubation, the slides were washed with PBS, and then streptavidin-horseradish peroxidase (HRP) was applied for 30 min. EnVision Detection System (DAKO, Jena, Germany) was applied, using 3,3′-diaminobenzidine as a substrate chromogen and counterstained with hematoxylin. Negative controls were treated by the same protocol, with a difference of using 1% BSA in PBS instead of a primary antibody. For positive external control for FKBPL, we used thyroid tissue.

### 2.3. Histopathological Evaluation

An analysis of immunohistochemical staining of FKBPL and VEGF-A was performed on 5 fields of magnification 100× and intensity was graded 0–3: 0—negative, 1—low, 2—moderate, 3—high intensity [11,12]. ERα expression was assessed through the Allred score, a well-established method for the quantification of ER in breast cancer, and suggested as a useful predictive factor in EC [19]. The Allred score was obtained as the sum of average intensity of stained nuclei, graded 0–3 (0—negative, 1—low, 2—moderate, 3—high intensity) and as the proportion of positive stained nuclei, graded 0–5 (0—negative, 1—less than 1%, 2—1–10%, 3—11–33%, 4—34–66% and 5—67–100% positive nuclei). The total value of the Allred score was in the range of 0–8, with an established cut-off value for positivity at a score value of 3. All slides had a blind analysis performed by two pathologists with 10 and 25 years of experience in the pathology of the female reproductive system (D.D.O. and D.M.O.).

### 2.4. Statistical Analysis

Numerical data are expressed as mean with standard deviation or as median with interquartile range. Categorical data are presented by absolute numbers with percentages and analyzed using a Chi-square test and Fisher exact test. For continuous variables, the Student t-test or the Mann–Whitney U test was used. The reliability of double-blinded readings was assessed by the Cronbach alpha coefficient. Measures of the diagnostic accuracy of FKBPL in discriminating between benign hyperplasia and endometrial carcinoma were determined by the receiver operating characteristic (ROC) analysis, and the cutoff level was determined. Sensitivity was defined as the % of patients with endometrial carcinoma who have no or low FKBPL expression (lower than the cut-off) (FKBPL no or low expression carcinoma patients/number of all carcinoma patients). Specificity was defined as the % of patients with hyperplasia who have FKBPL moderate or high expression levels (higher than the cut-off) (FKBPL moderate and high expression hyperplasia patients/number of hyperplasia patients). The positive predictive value was defined as the % of no or low FKBPL expression patients who have carcinoma (FKBPL no or low expression carcinoma patients/number of all FKBPL no or low expression patients). A negative predictive value was defined as the % of FKBPL moderate or high expression patients who have hyperplasia (FKBPL moderate or high expression hyperplasia patients/number of all FKBPL moderate or high expression patients). Correlations were examined by correlation coefficients according to the data scale used in the analyses (nominal by nominal, nominal by ordinal, and ordinal by scale). In all tests, *p* < 0.05 was considered statistically significant. Statistical analysis was performed using IBM SPSS statistical software (SPSS for Windows, release 25.0, SPSS, Chicago, IL, USA).

## 3. Results

### 3.1. Study Population

Clinical and histological data of cancer patients are presented in Table 1. The mean age of patients diagnosed with EC was 65.58 ± 8.59 years, and 45.23 ± 5.79 years for patients with BEH. Most patients with EEC were G1 and G2 (95.6%) and were diagnosed at an early-stage T1 (69.7%). EECs were predominantly limited to and did not extend beyond the uterus (89.9%).

### 3.2. Expression of FKBPL, VEGF-A and ERα in EEC and BEH

#### 3.2.1. Reliability of Double-Blind Reading

Chronbach’s alpha coefficient presented a high level of agreement between results obtained by a double-blind reading of immunohistochemical staining for FKBPL, ERα, and VEGF-A (Table 2).

#### 3.2.2. Difference of FKBPL Expression in EEC and BEH

There was a significant difference in FKBPL expression between EEC and BEH (*p* < 0.001), which demonstrated a decrease in intensity and loss of expression within the tissue sections with malignant changes of the endometrium (Table 3, Figure 1).

There was no significant correlation between FKBPL expression and the histological grade or clinical stage of the tumor or depth of myometrial invasion, or lymphovascular invasion (*p* > 0.05).

#### 3.2.3. Measures of Diagnostic Accuracy for FKBPL Expression

Immunohistochemical analysis showed a moderate to high intensity of FKBPL expression in 97.5% (*n* = 39) of the samples of BEH, and low or no expression in 93.3% (*n* = 83) of the cases of EEC. FKBPL staining showed a high positive predictive value (98.8%), and a high negative predictive value for malignant diagnosis (86.7%) (Table 4).

#### 3.2.4. Expression of ERα in EEC and BEH

ERα expression was found to be of moderate to high intensity in all samples of BEH, showing high values for the Allred score in 90% (*n* = 36) of the cases. ERα positivity in the EEC samples was determined to be of low intensity in 46.1% (*n* = 41), moderate intensity in 26.95% (*n* = 24), and high intensity in 26.95% (*n* = 24). In the group of patients with EEC, 78.7% (*n* = 70) showed expression of ERα in less than 66% of tumor glands, while 21.3% (*n* = 19) showed positive reaction in over 67% of tumor glands. High Allred score values were seen in 28.1% (*n* = 25), while moderate score values were observed in 68.6% (*n* = 61) of the EEC samples, whereas 2.2% (*n* = 2) showed low score values. Overall, there was a significant decrease (*p* < 0.001) in ERα expression between the EEC and BEH samples (Figure 2, Table 3).

The intensity of FKBPL expression, observed on the complete sample set from BEH and EEC patients, was in moderate positive correlation (*p* < 0.05) with the parameters of ERα expression (intensity, percentage and high Allred score values). The Allred score showed the strongest correlation.

#### 3.2.5. Expression of VEGF-A in EEC and BEH

Immunohistochemical expression of VEGF-A was not demonstrated in 95% (*n* = 38) of BEHs, whereas a low and moderate intensity of expression was determined in 42.7% (*n* = 38) of EECs (Figure 3, Table 3).

The intensity of FKBPL expression, observed on the complete sample set of BEHs and EECs, was in moderate negative correlation (*p* < 0.05) with the intensity of expression of VEGF-A.

## 4. Discussion

### 4.1. Expression of FKBPL in EEC and BEH

This study demonstrates the presence of low intensity or loss of expression of FKBPL in 93.3% of ECs, and a moderate to high intensity of expression in 97.5% of BEHs, suggesting that the intensity of FKBPL expression could be considered as a potential diagnostic biomarker in routine endometrial curettage examination with high positive and negative predictive value, differentiating between benign and malignant changes of the endometrium.

These findings are in accordance with studies analyzing the expression of FKBPL in a medium of healthy cell culture and in cell culture of BC, where FKBPL was detected in the medium of healthy cell lines and was absent in the medium of BC culture unless it was experimentally overexpressed. The study concluded that FKBPL is an endogenous secreted antiangiogenic protein, whose downregulation in neoplastic cells allows uncontrolled tumor growth potential. The same study showed that the administration of purified recombinant FKBPL and its peptide derivative, AD-01, inhibited the migration of tumor cells [20]. A similar study reported a low level of secretion of FKBPL in the BC cell line when compared with human microvascular endothelial cells (HMEC-1), and a decreased secretion of FKBPL by HMEC-1 when cultivated under hypoxic conditions. Part of this study examined tumor growth in a FKBPL knockdown mouse model, confirming increased vascular sprouting and tumor growth [21]. In our study, we showed no or low expression of FKBPL in EECs, which was substantially lower than in BEHs and in agreement with previously reported data. The lack of correlation of FKBPL with tumor grade, clinical stage, DMI, and LVI, might be due to limiting factors of the study, including that most of the EECs were of low grade (grade 1 and 2) and stage I, producing a group of biologically similar tumors. These limitations correspond to usual practice circumstances, since the vast majority of EEC cases are diagnosed at an early stage due to a typical symptom of abnormal uterine bleeding that warrants further examination of patients and timely diagnosis. The clinical and biological similarity of this experimental group is also influenced by the FIGO grading system, classifying ECs as: well, moderately, and poorly differentiated (grades 1, 2, and 3), which are grouped as low grade (grade 1 and 2) and high grade (grade 3) in cases of EEC [6]. In addition, this distinction is aligned to the traditional classification of ECs according to Bokhman as type I, low-grade endometrioid, dominantly estrogen-dependent, with more favorable prognosis, and type II, endometrial carcinomas of non-endometrioid morphology, including undifferentiated and dedifferentiated ECs [3,4,5].

### 4.2. Expression of ERα and FKBPL

Our findings showed a decrease in ERα expression in EECs compared with BEHs (*p* < 0.001). In addition, there was a moderate positive correlation between the intensity of the expression of FKBPL and ERα when BEHs and EECs were combined. These findings are interesting given that the previous reports showed FKBPL downregulating ERα expression in BC, leading to decreasing phosphorylation of ERα, and increasing sensitivity to tamoxifen; in addition, FKBPL expression was upregulated by estrogen treatment [14]. Therefore, FKBPL and ERα expression are expected to be negatively correlated. A possible explanation for these differences might be due to the fact that FKBPL expression is stimulated by estrogen, whereas in our study, most of our patients were of postmenopausal age, with likely lower circulating estrogen levels. Limitations of our study include the lack of data on the potential use of estrogen replacement therapy, body mass index, and the inability to stratify patients according to the low/high ERα expression in the EECs group due to the large percentage of ERα positive tumors. Further findings might be obtained from a larger group containing more ERα negative EECs. In addition, it has been reported that the effects of ERα antagonists, including tamoxifen, a known risk factor for the development of EEC, are tissue-specific, differing between EEC and BC by inhibiting the growth of BC cells and stimulating the growth of EECs [22]. Our findings correspond to two known dilemmas surrounding EECs. The first is that estrogen stimulation of the endometrium unopposed by progesterone plays an important role in the carcinogenesis of EEC; however, EECs predominantly occur in postmenopausal women who have lower levels of circulating estrogens. This is called “the endometrial carcinoma paradox”. The second dilemma to which our findings are aligned is the increased incidence of EECs in women treated with tamoxifen, which occurs in parallel to the inhibition of BC cells, despite the fact that both EEC and BC are highly estrogen-dependent tumors. Therefore, the role of ERα in EEC is still not fully understood and is somewhat contradictory. Hence, future studies should focus on elucidating this mechanism further.

### 4.3. Expression of VEGF-A and FKBPL

In this study, we showed an increase in VEGF-A expression in EECs compared with BEHs. In addition, when the complete sample set of BEHs and EECs was analyzed, we found a moderate negative correlation between the expression of FKBPL and VEGF-A (*p* < 0.05). Previous studies have shown that VEGF does not affect FKBPL expression, which was experimentally demonstrated ex vivo using an aortic ring assay following treatment with VEGF, where increased sprouting was indicative of increased angiogenesis from both wild-type and heterozygous FKBPL knockdown murine aortas, suggesting that FKBPL and VEGF affect angiogenesis through different pathways [21]. In light of these reports, the negative correlation that we have reported between FKBPL and VEGF-A might indicate an indirect interaction or two independent changes that complement each other as part of a pro-angiogenic switch in EEC. Also considering that previously published data relate both FKBPL and VEGF-A to estrogen and ERα [10,14], there could be a potential crosslink in the regulation of the balance between these two proteins with contrasting angiogenic effects. Several therapeutic approaches have been developed showing favorable results in blocking VEGF action, including blocking antibodies, decoy receptors, and small interfering RNA targeted at VEGF-A mRNA. However, anti-angiogenic therapy is still limited to small specific groups of patients, is associated with adverse effects, and many initially responsive patients develop resistance over time, creating a need for further understanding of the mechanisms and factors involved in the angiogenic process of EC [9,10]. An FKBPL peptide derivative, ALM-201, completed a phase I first-in-human clinical trial for ovarian cancer and other advanced solid tumors, which included two patients with EC in the safety trial group. Subsequently, ALM-201 was designated as an orphan drug for ovarian cancer by the Food and Drug Administration [23]. Further studies investigating FKBPL and its therapeutic peptide-derivative use in patients with EEC are needed in order to confirm the results obtained in our study. Considering that EEC is an evolving disease with a heterogeneous presentation and unelucidated mechanisms of carcinogenesis, there is a need for the pursuit of novel diagnostic and therapeutic approaches, which could include FKBPL-based markers and therapies that harness this emerging mechanism in EECs.

## 5. Conclusions

To our knowledge, we present the first comprehensive study on the immunohistochemical expression of FKBPL in EEC, showing a significant decrease in FKBPL in comparison to BEH with a high predictive value of malignancy. In addition, the study gives insight into the correlation between FKBPL expression and ERα as a known effector of endometrial proliferation, and VEGF-A as one of the central pro-angiogenic factors and target proteins for anti-angiogenic therapy in EEC patients.

## Figures and Tables

**Figure 1 medicina-58-01330-f001:**
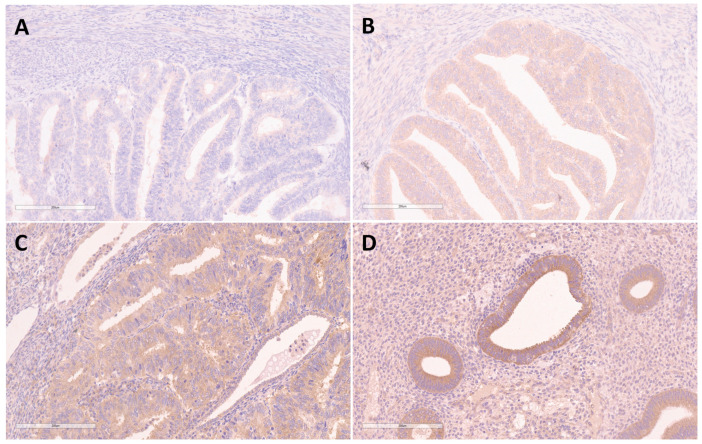
Immunohistochemical staining for FKBPL in endometrioid endometrial carcinoma showing no expression—0 (**A**), low expression—1 (**B**), moderate expression—2 (**C**), and immunohistochemical staining for FKBPL in benign endometrial hyperplasia showing high expression—3 (**D**). Magnification ×200.

**Figure 2 medicina-58-01330-f002:**
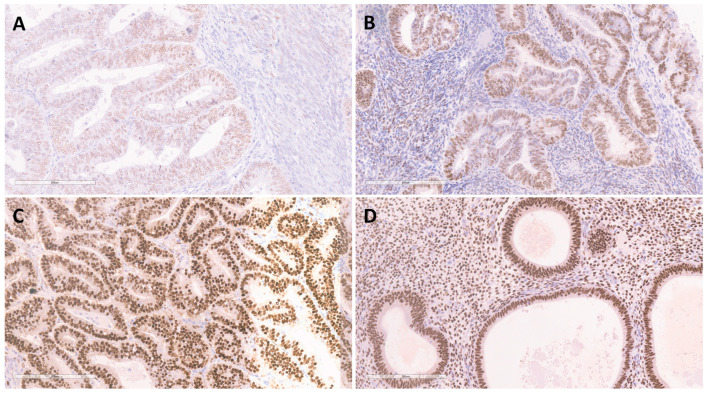
Immunohistochemical staining for ERα in endometrioid endometrial carcinoma showing low expression—1 (**A**), moderate expression—2 (**B**), high expression—3 (**C**), and immunohistochemical staining for ERα in benign endometrial hyperplasia showing high expression—3 (**D**). Magnification ×200.

**Figure 3 medicina-58-01330-f003:**
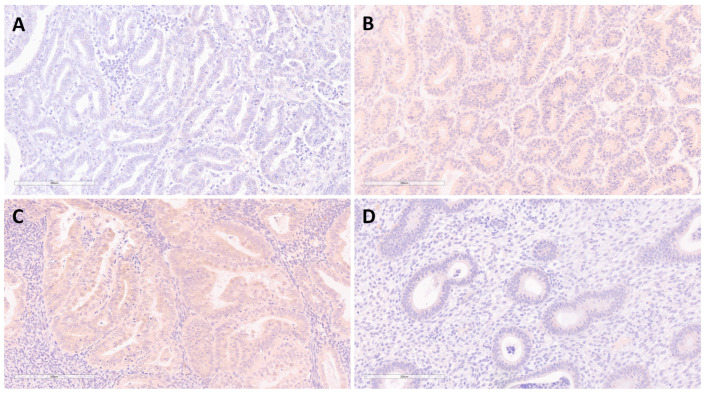
Immunohistochemical staining for VEGF-A in endometrioid endometrial carcinoma showing no expression—0 (**A**), low expression—1 (**B**), moderate expression—2 (**C**), and immunohistochemical staining for VEGF-A in benign endometrial hyperplasia showing no expression—0 (**D**). Magnification ×200.

**Table 1 medicina-58-01330-t001:** Clinical and histological data of cancer patients.

Variable		EEC (*n* = 89)
Grade (%)	G1	40.4 (*n* = 36)
	G2	55.1 (*n* = 49)
	G3	4.5 (*n* = 4)
TNM stage (%)	T1A	33.7 (*n* = 30)
	T1B	36 (*n* = 32)
	T2	20.2 (*n* = 18)
	T3	10.1 (*n* = 9)
FIGO stage (%)	IA	31.5 (*n* = 28)
	IB	36 (*n* = 32)
	II	20.2 (*n* = 18)
	III	11.2 (*n* = 10)
	IV	1.1 (*n* = 1)
Myometrial invasion depth (%)	Less than half	37.1 (*n* = 33)
	One half or more	62 (*n* = 56)
Lymphovascular invasion (%)	Yes	33.7 (*n* = 30)
	No	66.3 (*n* = 59)

Tumor-Node-Metastases (TNM)–based staging, International Federation of Gynecology and Obstetrics (FIGO) staging system.

**Table 2 medicina-58-01330-t002:** Reliability of double-blind reading.

Variable	Cronbach’s Alpha	Internal Consistency
FKBPL	0.994	Excellent
ERα (intensity)	0.997	Excellent
ERα (percentage)	0.987	Excellent
Allred score	0.996	Excellent
VEGF-A	0.970	Excellent

FK506-binding protein-like (FKBPL), vascular endothelial growth factor-A (VEGF-A), estrogen receptor alpha (ERα).

**Table 3 medicina-58-01330-t003:** Expression of FKBPL, VEGF-A, and ERα in BEH and EEC.

Variable			Benign Endometrial Hyperplasia *n* = 40	Endometrioid Endometrial Carcinoma *n* = 89	Significance (*p*)
FKBPL	0/1		2.5% (*n* = 1)	93.3% (*n* = 83)	<0.001
	2/3		97.5% (*n* = 39)	6.7% (*n* = 6)	
ERα	Intensity	1	0% (*n* = 0)	46.1% (*n* = 41)	<0.001
		2/3	100% (*n* = 40)	53.9% (*n* = 48)	
	Percentage	<5 (≤66%)	20.0 (*n* = 8)	78.7% (*n* = 70)	<0.001
		5 (67–100%)	80.0 (*n* = 32)	21.3% (*n* = 19)	
	Allred Score	<7	10% (*n* = 4)	71.9% (*n* = 64) *	<0.001
		7/8	90% (*n* = 36)	28.1% (*n* = 25)	
VEGF-A	0		95% (*n* = 38)	57.3% (*n* = 51)	<0.001
	1/2		5% (*n* = 2)	42.7% (*n* = 38)	

Benign endometrial hyperplasia (BEH), endometrial endometrioid carcinoma (EEC). * 1 sample of endometrioid endometrial carcinoma had Allred score <3.

**Table 4 medicina-58-01330-t004:** Measures of diagnostic accuracy for FKBPL expression.

Variable	Sensitivity	Positive Predictive Value	Specificity	Negative Predictive Value	Accuracy
FKBPL	93.3%	98.8%	97.5%	86.7%	94.6%

## Data Availability

Not applicable.

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
