# Peer review of "Loss of Expression of Antiangiogenic Protein FKBPL in Endometrioid Endometrial Carcinoma: Implications for Clinical Practice"

_medicina, 2022, doi:10.3390/medicina58101330_

Round 1
Reviewer 1 Report
Proofreading is required on lines 120, 131, 165, 168, 170, 173, 232, 234.
Table 1 should be corrected as the format is not acceptable
Section 3.2 should either be expanded or assigned to another section. The necessity of the internal consistency of Table 2 is questionable
Author Response
We kindly thank you for the revision of our paper and we are appreciative of the time and effort that you dedicated to providing valuable and extensive feedback on our manuscript. Also, we are grateful for the insightful comments on, that will add valuable improvements to our paper. We have incorporated all of the suggestions made. Those changes are highlighted within the manuscript. Please see below, our point-by-point response to the comments and concerns.
- Comment from Reviewer 1:’’Proofreading is required on lines 120, 131, 165, 168, 170, 173, 232, 234’’
Author response: Thank you kindly for pointing this out. We have proofread all the lines pointed and corrected all typographic mistakes we observed.
- Comment from Reviewer 1: ’’Table 1 should be corrected as the format is not acceptable’’
Author response: Thank you kindly for pointing this out. We have changed the format of Table 1. by rearranging results on myometrial invasion depth and lymphovascular invasion, hopefully making Table 1. more transparent. Also, with the same goal, we removed data on benign endometrial hyperplasia patients, which are already described in the materials and method section, and reduced endometrial endometrioid carcinoma title to EEC.
- Comment from Reviewer 1: ’’Section 3.2 should either be expanded or assigned to another section. The necessity of the internal consistency of Table 2 is questionable’’
Author response: Thank you kindly for pointing this out. We have assigned the section ’’Reliability of double blind reading’’ to the section ’’Expression of FKBPL, VEGF-A, and ERα in EEC and BEH’’. We wanted to keep Table 2. wishing to show the consistency of our protocol especially concerning FKBPL staining because immunohistochemical staining reports for this antibody in endometrial carcinomas are extremely rare.
Reviewer 2 Report
The aim of this study was to 110 examine the expression of FKBPL and its correlation with the expression of VEGFR and 111 ERα in BEH and EEC.
In this, the authors have succeeded in writing a good "technique" paper.
Suggestions:
Introduction is too long for a short technique paper.
I perceive an underline assumption by the authors that highly angiogenic tumors should be aggressive and should have worse prognosis in comparison with less angiogenic tumors when the say...."It is shown that estradiol (E2) stimulates the proliferation and migration of endometrial cancer stem cells (CSC) through ERα. Paradoxically, the presence of ERα positivity in EECs is shown to be a positive prognostic marker, as opposed to ERα negative EECs that are more aggressive tumors with poorer prognoses."...In fact, most epithelial exophytic tumors, since they raise above surface, can do so because of vascular proliferation. Proliferative stroma lacks function lymphatics whereas mature lymphatics are present in the normal tissue at the infiltrative margin (base) of the tumor. lack of angiogenesis compels tumor to grow where host's microvasculature is available and hypoxic environment causes phenotypic changes in the neoplastic cells leading to loss of adhesion molecules, de-differentiation and MELF like changes,. Such cells can easily be "sucked" into the terminal lymphatics, LVSI. Afterall, all the normal day to day interstitial tissue debris is cleared by the terminal lymphatics without requiring any special passports to get in the lymphatics!
......line 67..Growth of EEC, before the tumor exceeds a volume of 2mm3, is independent of vascularisation,......please provide a reference, since you have given a specific data.
One of the common deficiency in publications related to molecular prognostic markers have been almost a complete lack of quantitative co-relation with the known prognostic markers. In endometrial cancer, that will be say LVSI, nodes of FFS and OS.
Line 203.....
There was no significant correlation between FKBPL expression and histological grade or clinical stage of the tumor or depth of myometrial invasion, or lymphovascular 204 invasion (p>0.05). This should be discussed / explained.
Author Response
We kindly thank you for the revision of our paper and we are appreciative of the time and effort that you dedicated to providing valuable and extensive feedback on our manuscript. Also, we are grateful for the insightful comments on, that will add valuable improvements to our paper. We have incorporated all of the suggestions made. Those changes are highlighted within the manuscript. Please see below, our point-by-point response to the comments and concerns.
- Comment from Reviewer 2: ’’ The aim of this study was to 110 examine the expression of FKBPL and its correlation with the expression of VEGFR and 111 ERα in BEH and EEC.
In this, the authors have succeeded in writing a good "technique" paper.’’
Author response: We kindly thank you for your assessment. We truly appreciate it.
- Comment from Reviewer 2: ’’ Introduction is too long for a short technique paper.’’
Author response: Thank you kindly for pointing this out. We have reduced the introduction text as much as possible trying to maintain clarity. The word count is reduced from approximately 860 to 790 words.
- Comment from Reviewer 2: ’’ I perceive an underline assumption by the authors that highly angiogenic tumors should be aggressive and should have worse prognosis in comparison with less angiogenic tumors when the say...."It is shown that estradiol (E2) stimulates the proliferation and migration of endometrial cancer stem cells (CSC) through ERα. Paradoxically, the presence of ERα positivity in EECs is shown to be a positive prognostic marker, as opposed to ERα negative EECs that are more aggressive tumors with poorer prognoses."...In fact, most epithelial exophytic tumors, since they raise above surface, can do so because of vascular proliferation. Proliferative stroma lacks function lymphatics whereas mature lymphatics are present in the normal tissue at the infiltrative margin (base) of the tumor. lack of angiogenesis compels tumor to grow where host's microvasculature is available and hypoxic environment causes phenotypic changes in the neoplastic cells leading to loss of adhesion molecules, de-differentiation and MELF like changes,. Such cells can easily be "sucked" into the terminal lymphatics, LVSI. Afterall, all the normal day to day interstitial tissue debris is cleared by the terminal lymphatics without requiring any special passports to get in the lymphatics!’’
Author response: Thank you kindly for pointing this out. In the pointed section we have discussed the effects of estradiol on cancer cells of endometrial carcinoma, also the findings on the presence and absence of specific estrogen receptor alpha expression in tumor cells as a prognostic factor, without mentioning endothelial or vascular changes. We appreciate the extensive insights in this topic, and tending to incorporate them and to clarify this part of the text we added the explanatory note saying that worse prognosis in ERα negative EECs ’’ might be due to dedifferentiation of tumor cells’’.
- Comment from Reviewer 2: ’’ ......line 67..Growth of EEC, before the tumor exceeds a volume of 2mm3, is independent of vascularisation,......please provide a reference, since you have given a specific data.’’
Author response: Thank you kindly for pointing this out. The data on this subject is the part of the paper under reference number 9 which we used at the end of the mentioned paragraph. We have now added the number for that reference at the end of the sentence you mentioned to more clearly address it.
- Comment from Reviewer 2: ’’ One of the common deficiency in publications related to molecular prognostic markers have been almost a complete lack of quantitative co-relation with the known prognostic markers. In endometrial cancer, that will be say LVSI, nodes of FFS and OS.’’
Author response: Thank you kindly for pointing this out. We agree with your constatation and we wanted to include as many prognostic markers as possible. Sadly the data on FFS and OS was not available to us in patients documentation mostly due to dispersion of patients to regional oncologic centers for further treatment, lack of follow-up in the center where initial diagnostics were performed, and lack of communication and connection between informatic data systems of different centers in our country.
- Comment from Reviewer 2: ’’ Line 203.....There was no significant correlation between FKBPL expression and histological grade or clinical stage of the tumor or depth of myometrial invasion, or lymphovascular 204 invasion (p>0.05). This should be discussed / explained.’’
Author response: Thank you kindly for pointing this out. The lack of correlation between FKBPL and available prognostic markers that we analyzed was addressed in discussion section 4.1. ’’Expression of FKBPL in EEC and BEH’’. We addressed these findings as potentially due to limiting factors of the study, including that most of our patients had low-grade (grade 1 and 2) EECs and were in stage I, making the group of biologically similar tumors.